# Antimicrobial Effects of Tetraspanin CD9 Peptide against Microbiota Causing Armpit Malodour

**DOI:** 10.3390/antibiotics12020271

**Published:** 2023-01-29

**Authors:** Hassanain Al-Talib, Marwa Hasan Abdulwahab, Khairiyah Murad, Nur Deanna Amiruddin, Normi Ngah Mohamed

**Affiliations:** 1Department of Medical Microbiology and Parasitology, Faculty of Medicine, Universiti Teknologi MARA (UiTM), Sungai Buloh 47000, Selangor, Malaysia; 2Microbiology Department, Collage of Science, Tikrit University, Tikrit 34012, Iraq; 3Institute for Medical Molecular Biotechnology, Universiti Teknologi MARA (UiTM), Sungai Buloh 47000, Selangor, Malaysia

**Keywords:** tetraspanin, synthetic peptides, CD9, antibacterial, anti-biofilm

## Abstract

Synthetic peptides, including tetraspanin CD9 peptides, are increasingly coming into focus as new treatment strategies against various organisms, including bacteria, that cause underarm odour. The use of deodorants and antiperspirants is associated with side effects. Therefore, it is critical to find an alternative therapeutic approach to combat underarm odour. The aim of this study is to investigate the antibacterial effect of tetraspanin CD9 peptides against the skin microbiota that cause malodour in the underarms. The antimicrobial activity of CD9 peptides against *Micrococcus luteus* (*M. luteus*), *Bacillus subtilis* (*B. subtilis*), *Staphylococcus epidermidis* (*S. epidermidis*), and *Corynebacterium xerosis* (*C. xerosis*) was investigated by the disc diffusion method. Minimum inhibitory concentration (MIC) and minimum bactericidal concentration (MBC) were determined by broth microdilution assays using CD9 peptide concentrations ranging from 1 mg/mL to 0.0078 mg/mL. In addition, the anti-biofilm activity of the CD9 peptides was determined. The CD9 peptides showed different antibacterial activity with an inhibition zone of 7.67, 9.67, 7.00, and 6.00 mm for *S. epidermidis*, *M. luteus*, *C. xerosis*, and *B. subtilis*, respectively. All bacteria had the same MBC value of 1 mg/mL. A high MIC of CD9 peptides was observed for *S. epidermidis* and *M. luteus* at 0.5 mg/mL. The MIC values of *B. subtilis* and *C. xerosis* were 0.125 mg/mL and 0.25 mg/mL, respectively. CD9 peptides significantly inhibited biofilm development of *S. epidermidis*, *B. subtilis*, and *C. xerosis* isolates. The CD9 tetraspanin peptide has excellent antibacterial activity against bacteria that cause underarm odour. Therefore, the CD9 tetraspanin peptide is a promising alternative to deodorants and antiperspirants to combat commensal bacteria of the skin that cause underarm odour.

## 1. Introduction

Unpleasant underarm odour is due to microbial biotransformation of odourless and sterile secretions into volatile odour molecules caused by enzymes released by the microbiome in sweat glands, pores, and hair follicles [1]. In the armpit, there are glandular fields with discrete bacterial colonies, which ideally live in a sweaty and humid environment where oily and odourless fluids are the main source of nourishment [2]. The bacterial flora of the axilla proved to be a stable mixture of Gram-positive bacteria, including *Micrococcus luteus* (*M. luteus*), *Bacillus subtilis* (*B. subtilis*), *Staphylococcus epidermidis* (*S. epidermidis*), and *Corynebacterium xerosis* (*C. xerosis*) [3]. Odourless secretions are secreted by eccrine, sebaceous, and apocrine glands to regulate body temperature [4]. In contrast to primary bacterial catabolism, the bacterial flora of the axilla undergoes processes such as steroid biotransformation of long-chain fatty acids to release short volatile branched fatty acids [5]. People tend to use various topical deodorants and antiperspirants to minimize the spread of underarm odour. Deodorants work by lowering the bacterial count and masking the bad odour with fragrance molecules such as ethanol or an antibacterial agent that kills the bacteria [6]. On the other hand, antiperspirants contain aluminium chloride and reduce moisture in the armpits by temporarily clogging the skin pores leading to the sweat glands [7]. One thing to consider with these topical treatments is that they will create space for new species, even though they reduce the current bacterial density [2]. Tetraspanins are small transmembrane proteins with four transmembrane domains that form a small intracellular loop and two extracellular loops, EC1 and EC2, where most protein–protein interactions occur, and are also a common target for anti-tetraspanin monoclonal antibodies [8]. They ought to create a specialized tetraspanin-enriched microdomain (TEM) by forming promiscuous bonds with each other and with other cell components, such as structural proteins or signalling molecules [9]. 

TEM is essential for cell adhesion and fusion, endocytosis, and membrane trafficking, but also forms an entry gate for pathogens [10]. In other words, tetraspanins are mediator molecules for bacterial adhesion with host cells during bacterial infection [11]. Therefore, tetraspanins have the potential to fight infections caused by microbes. 

Recent data suggest that tetraspanins are abundantly expressed by human cells that regulate the binding of bacterial toxins to host cells, which then promotes infections [12]. One of the tetraspanins is known as cluster of differentiation 9 (CD9) or (TSPAN29), and is involved in numerous cellular processes, such as cell–cell contact, cell–extracellular matrix interaction, signalling, membrane fusion, inflammation, proliferation, and differentiation [13]. Researchers have demonstrated that the recombinant EC2 domain in CD9 tetraspanin peptides reduces the attachment of bacteria, such as *Neisseria meningitidis* and *Salmonella enterica,* to epithelial cells by this peptide. In addition, anti-tetraspanin antibodies, such as CD9-EC2 peptides, can reduce the attachment of P. aeruginosa to keratinocytes and epithelial cells labelled with green fluorescent protein [14]. We hypothesize that the newly synthesized CD9 peptides are able to inhibit the bacterial axillary flora responsible for underarm odour. Therefore, this study aims to investigate the inhibitory effect of tetraspanin CD9 peptides against the bacterial flora responsible for unpleasant underarm odour.

## 2. Results

The antibacterial effect of CD9 tetraspanin peptides on the four most common axillary flora bacteria responsible for axillary odour was evaluated. A susceptibility test using the disc diffusion method was performed on axillary bacterial flora (Figure 1). This study showed that *M. luteus* was sensitive to CD9 tetraspanin peptides, as it showed the most distinct zone of inhibition (mean 9.67 mm ± 4.04). On the other hand, CD9 tetraspanin peptides had no inhibitory effect on *B. subtilis* (mean 6.00 mm ± 0). In contrast, CD9 tetraspanin peptides exhibited moderate inhibitory activity against both *S. epidermidis* (mean 7.67 mm ± 2.89) and *C. xerosis* (mean 7.00 mm ± 2.00), as shown in Table 1. For MIC, experiments were performed in triplicate, and absorbance values were closely compared with the opacity of wells observed by the naked eye. The MIC is a well that has no opacity or cloudiness. This study demonstrated that only *B. subtilis* can be stably inhibited at a low concentration of CD9 tetraspanin peptides of 0.125 mg/mL. However, the growth of *S. epidermidis* and *M. luteus* can only be inhibited with a high peptide concentration of 0.5 mg/mL Table 2. *C. xerosis* only showed no turbidity at a CD9 tetraspanin peptide concentration of 0.25 mg/mL. As for MBC, all four bacterial isolates showed no growth at the highest peptide concentration of 1 mg/mL (Figure 2). 

Inhibition of bacterial biofilm by CD9 peptide showed that *S. epidermidis* was excellently inhibited, as the absorbance values decreased significantly from (0.120 ± 0.004) to (0.065 ± 0.003) in the presence of the CD9 tetraspanin peptide (Table 3). The optical density values of *B. subtilis* and *C. xerosis* (0.090 ± 0.003) and (0.083 ± 0.002) were significantly reduced to (0.062 ± 0.003) and (0.061 ± 0.004), respectively, compared with the CD9 tetraspanin-peptide-treated group. Conversely, *M. luteus* was resistant to the CD9 tetraspanin peptide and, therefore, showed a high absorbance value (0.080 ± 0.012) in the treated well, whereas it showed a lower value (0.073 ± 0.002) in the untreated well (Figure 3 and Figure 4). The greatest biofilm inhibition was observed with S. *epidermidis* 45.8% followed by *B. subtilis* and *C. xerosis* 31.11% and 26.51%, respectively. 

The data above are based on the mean values of minimal inhibitory concentration (MIC) and minimal bactericidal concentration (MBC) in unit mg/mL, with naked-eye observation.

## 3. Discussion

Human underarm odour is a frustrating and bothersome health problem and has led to a bad perception of individuals who sweat uncontrollably. It is not only unpleasant, but can also cause psychological disorders if left untreated [2]. In some cases, body odour may develop due to poor hygiene and poor metabolism associated with daily diet. Therefore, researchers have pointed out that it is urgent to combat this problem through the use of topical treatments, such as deodorants and antiperspirants, that contain antibacterial compounds, such as alum and triclosan [15]. Topical underarm treatments have been associated with several health problems, especially in women who apply deodorant or antiperspirant directly, which can cause side effects, such as skin irritation and allergic contact dermatitis [16]. The focus of this study is on the antimicrobial inhibitory effect of CD9 tetraspanin peptides against the axillary microbiota as an alternative to antiperspirants whose ingredients contain aluminium, which is known to cause anaphylaxis, Alzheimer’s disease, and Parkinson’s disease, especially in immunocompromised individuals [17,18]. In this study, CD9 peptides were reported to have an inhibitory effect against *S. epidermidis,* with a mean zone of inhibition of 7.67 mm. Our result is similar to a previous study by Qin et al. (2006), which found a significant inhibitory effect of synthetic peptides against *S. epidermidis* [19]. However, the MIC and MBC of the CD9 tetraspanin peptide against *S. epidermidis* were somewhat high at 0.5 and 1 mg/mL, respectively. With these values, a higher dose of CD9 peptides should be used as an adjunct in hygiene products but can still be safely administered, especially if the designed peptide is less than 100 amino acids, according to the FDA’s updated definition of chemically synthesized polypeptides with more than 40 amino acids but less than 100 [20]. In addition, a wide range of synthetic peptides have been approved by the FDA for the treatment of various diseases, such as cancer, metabolic diseases, and infectious diseases. Because peptides are more effective, safer, and tolerated by humans and can penetrate cell membranes, synthetic peptides have emerged as promising drug candidates [20]. Synthetic peptides are currently considered suitable by the pharmaceutical industry because they are easily degraded in the human body, unlike other antibiotics that could produce potentially harmful metabolites [21,22]. This study showed that the CD9 tetraspanin peptide had no inhibitory effect on *B. subtilis* with an inhibition zone (6.00 mm ± 0). This may be due to the presence of the gene encoding of ABC-F-type ribosomal protection protein Lsa(B), whose protein protects the ribosome from the action of antibiotics [23]. *Bacillus Subtilis* is unique in that it has the ability to produce more than 24 types of antibacterial molecules [24], which consists of five signal peptidase genes in antimicrobial peptides, including cellulase, glucanase, chitinase, and protease [25]. Its defence mechanism combats other microbiota in the armpits, especially an odour-producing microbe, thus, reducing the bad odour in the armpits. In addition, *B. subtilis* is considered a bacterium that is generally recognized as safe. Therefore, it is used in many commercial products as a probiotic for human cosmetics. The MIC and MBC of the CD9 tetraspanin peptide against *B. subtilis* were 0.125 and 1 mg/mL, respectively. The current innovation in synthetic tactics provides more advantages for peptides to achieve the desired properties. In addition, synthetic peptides showed excellent antimicrobial activity and were relatively inexpensive, making them favourable candidates. Moreover, synthetic peptides showed almost no or limited development of resistance in various bacteria [26]. This makes them very promising compared to antibiotics, which develop resistance relatively quickly. The lack of this resistance development in microbes can be attributed to the different mode of action of synthetic peptides against bacteria compared to the standard modes of action of antibiotics [22]. 

*M. luteus* is a harmless saprophyte but can turn into an opportunistic species in the immunocompromised group with acute leukaemia and pneumonia [27]. This bacterium is known to cause odour nuisance in humans by breaking down the sweat components into odour molecules, thus, conquering a large space on the human skin [28]. The current study showed that the CD9 tetraspanin peptide had antimicrobial activity against *M. luteus,* with a mean zone of inhibition of 9.67 mm ± 4.04. Similarly, *M. luteus* is endowed with an inducible plasmid-borne antibiotic-resistance gene, pMEC2, which makes it resistant to erythromycin, other macrolides, and lincomycin [29]. The MIC and MBC of the CD9 tetraspanin peptide against *M. luteus* were 0.5 and 1 mg/mL, respectively. However, the antimicrobial activity of the tetraspanin peptide could be improved by adding a cofactor or substituting certain amino acids, as demonstrated for negatively charged peptides, as previously recommended by Offret et al. [30]. A previous study also demonstrated a lower MIC of the synthetic peptide against *M. luteus* (64 µg/mL), which could be due to variations in the techniques or concentration of the bacterial broth [31]. 

Moreover, *C. xerosis* results in the release of pungent axillary odour, which is strongly associated with the density of live aerobic coryneform species in the axillary microbiome flora. This study demonstrated a moderate inhibitory effect of the CD9 tetraspanin peptide against *C. xerosis* with mean inhibition zone (7.00 mm ± 2.00). The MIC and MBC of the CD9 tetraspanin peptide against *C. xerosis* were 0.25 and 1 mg/mL, respectively, consistent with previous results from Lui et al. (2003), who found that 10 synthetic peptides had an MIC range from 0.2 µM to 0.5 µM [32]. In a previous report by Hahne et al. (2018), *C. xerosis* was shown to be resistant to three different antimicrobial classes, i.e., multidrug resistant, as defined by the CLSI [33]; therefore, the CD9 tetraspanin peptide will have a chance to replace antimicrobials.

Biofilm formation, especially in multidrug-resistant bacteria, leads to serious life-threatening infections [34]. A previous study concluded that the sesC gene may encode an essential function in *S. epidermidis* biofilm formation [35]. Our study showed that the CD9 tetraspanin peptide could significantly reduce the primary attachment of *S. epidermidis* to polystyrene plates and inhibit biofilm formation by *S. epidermidis* compared with untreated plates (*p* value < 0.05). The inhibition of biofilm formation by the tetraspanin peptide could be due to the cross-reaction of the peptide with other surface-exposed *S. epidermidis* or to the absence of the sesC gene. In this study, we found that biofilm formation of *B. subtilis* and *C. xerosis* was significantly inhibited by the CD9 tetraspanin peptide. In addition, we found that the CD9 tetraspanin peptide had no significant effect on the biofilm formation of *M. luteus*. Our result differs from a previous study in which a higher peptide concentration of 4 µg was used [36]. The variation in inhibition of biofilm formation was dependent on peptide concentration and strain sensitivity. Therefore, higher peptide concentrations were required to inhibit *M. luteus* biofilm formation.

## 4. Materials and Methods

### 4.1. Bacterial Strains

In this study, the four most common bacterial strains responsible for producing foul odour in the human axilla were investigated: *M. luteus*, *B. subtilis*, *S. epidermidis*, and *C. xerosis* [37]. Bacterial strains used in this study were *Micrococcus luteus* (ATCC 49732), *Staphylococcus epidermidis* (ATCC 14990), *Corynebacterium xerosis* (ATCC BAA-1293), and *Bacilus subtilis* (ATCC 19659). These bacteria were ordered from a selected supplier. All bacterial strains were grown on blood agar and incubated at 37 °C for 24 h. For analysis, the bacterial suspension was prepared overnight, and its turbidity was adjusted to 0.5 McFarland scale by inoculating a few bacterial colonies from the blood agar in Luria-Bertani broth (LB broth).

### 4.2. CD9 Tetraspanin Peptide

The CD9 tetraspanin peptide was used primarily to test its ability to inhibit the bacterial activity of the bacteria used in this project. CD9, also known as TSPAN29, is derived from the human species. The EC2 tetraspanin CD9-derived peptide was purchased from (GenScript, Piscataway, NJ, USA) in lyophilized form with a molecular weight of 2195.44 g/mol. The peptide sequence was: EPQRETLKAIHYALN.

The CD9 peptide was prepared by dissolving 1 mg in 1 mL of dimethyl sulfoxide (DMSO). A neon pink-purple CD9 peptide solution with a concentration of 1 mg/mL was stored at 4 °C for short-term use or at −20 °C for long-term use. In addition, the peptide was serially diluted with sterile water to obtain concentrations from 1 mg/mL to 0.002 mg/mL.

### 4.3. Antimicrobial Susceptibility of Tetraspanin CD9 Peptide against Armpit Bacterial Flora

The inhibitory effect of the CD9 peptide on bacteria was determined by a disc diffusion assay. In this assay, 100 μL of a standardized 0.5 McFarland bacterial suspension was inoculated overnight with a sterile swab on Mueller-Hinton agar (MHA). A 6 mm blank disc containing 20 μL CD9 peptide with concentration of 1 mg/mL and a standard antibiotic disc, 30 μg vancomycin (Oxoid, Basingstoke, UK), was placed and gently pressed onto the surface of the MHA agar with forceps. The agar plate was then incubated at 37 °C for 24 h, and the zone of inhibition around the discs was measured with a ruler and expressed in units of “mm.” The measurement of the zone of inhibition around the blank antibiotic disc concentrated with CD9 peptide was compared with the zone of inhibition around the positive control.

### 4.4. Minimal Inhibitory Concentration (MIC)

The minimum inhibitory concentration was determined to be the lowest concentration of CD9 tetraspanin peptide that inhibits the growth of *M. luteus*, *S. epidermidis*, *B. subtilis*, and *C. xerosis*. First, 50 μL of a serial dilution of CD9 tetraspanin peptide (1 mg/mL to 0.002 mg/mL) was added to the wells, followed by 50 μL of a 0.5 McFarland bacterial suspension overnight for each bacterium. A positive control (bacterial suspension and LB broth) and a negative control (LB broth and distilled water) were also prepared for comparison. Then, the plate was incubated overnight at 37 °C. The following day, absorbance at 595 nm was measured and tabulated, while turbidity of the solution in each well was observed by the naked eye. The MIC was determined in the first well, which showed no bacterial growth or turbidity.

### 4.5. Minimal Bactericidal Concentration (MBC)

The minimum bactericidal concentration was determined to be the lowest concentration of CD9 tetraspanin peptide required to kill bacterial flora in the axilla. MBC was determined by diluting 10 μL of the bacterial broth from the MIC in 90 μL of sterile distilled water. The diluted broth was then spread evenly on blood agar using a cell spreader. The agar plates were then incubated overnight at 37 °C. The next day, all plates and bacterial growth were examined and recorded. The MBC value was marked on the agar plate that showed no growth of bacterial colonies.

### 4.6. Bacterial Biofilm Inhibition by CD9 Tetraspanin Peptide

Approximately 50 μL of overnight bacterial suspension at 0.5 McFarland was added together with 50 μL of CD9 tetraspanin (0.5 mg/mL) peptide into the wells of the 96-well microtiter flat-bottom base plate. For comparison, an untreated well was also prepared for each bacterial strain by mixing 50 μL of sterile water with 50 μL of the overnight bacterial suspension. In addition, LB broth was added to a separate well as a negative control. After completion, the plate was incubated at 37 °C for 24 h. After incubation, the plate was washed three times with sterile water. The wells were then filled with 200 μL of 100% methanol and incubated for 15 min at room temperature. This was carried out to remove all non-adherent bacteria in the wells. We continued the washing step, as above, three times to rinse off the non-adherent bacteria. After washing, the plate was inverted and air dried at room temperature for 30 min. The dried plate was filled with 125 μL of 0.1% crystal violet for staining. Then, the plate was washed three times to remove the stain and leave only the biofilm adhered to the wells. The plate was then air-dried and inverted for 30 min. To dissolve the biofilm, 200 μL of 30% glacial acetic acid solution was added to each well for 10 min. Then, 125 μL of the well contents was transferred to a new 96-well microtiter plate. Finally, absorbance was measured at 595 nm. The results were tabulated, and the experiment was repeated three times independently.

### 4.7. Statistical Analysis

Each experiment was performed in 3 replicates with 3 independent trials. The data were entered and analysed using Statistical Package for Social Sciences version 20.0 (SPSS Inc., Chicago, IL, USA). Tabulated data are the means ± standard deviation of the three replicate trials.

## 5. Conclusions

The CD9 tetraspanin peptide has excellent antibacterial activity against *S. epidermidis*, *M. luteus*, and *C. xerosis* and weak activity against *B. subtilis*. However, it has a strong effect against biofilm formation on *S. epidermidis*, *B. subtilis,* and *C. xerosis* and, in turn, prevents bacterial cell adhesion. From the combination of the direct bacterial inhibitory effect and the inhibition of biofilm formation by bacteria, we can conclude that the CD9 tetraspanin peptide is a promising alternative to combat commensal bacteria of the skin that cause axillary malodour. Therefore, a more comprehensive in vivo study should be conducted to test its reliability as an alternative topical treatment in deodorants and antiperspirants against underarm bacterial flora.

## Figures and Tables

**Figure 1 antibiotics-12-00271-f001:**
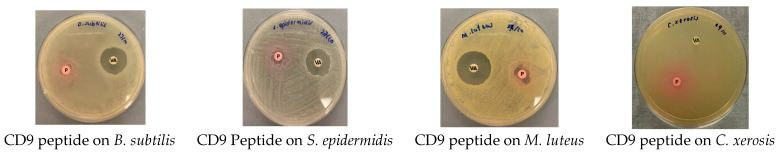
Inhibitory effects of CD9 tetraspanin peptide on armpit bacterial flora. VA: Vancomycin, P: CD9 Peptide.

**Figure 2 antibiotics-12-00271-f002:**
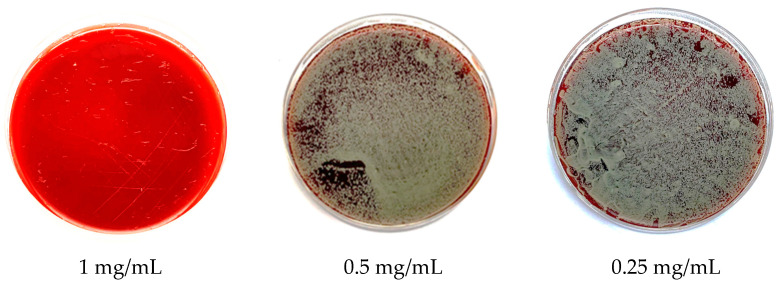
MBC of CD9 tetraspanin peptide against *M. luteus* on blood agar.

**Figure 3 antibiotics-12-00271-f003:**
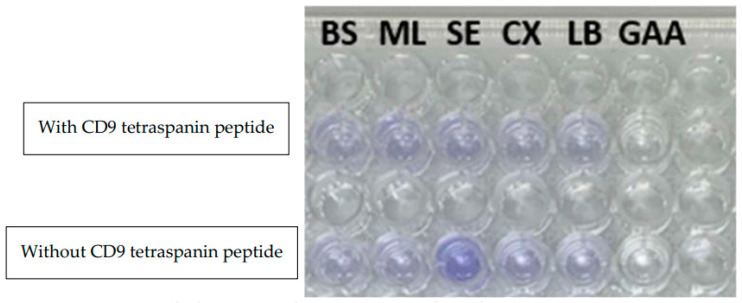
Inhibition of bacterial biofilm by CD9 tetraspanin peptide against axillary microbiota. BS: *B. subtilis*, ML: *M. luteus*, SE: *S. epidermidis*, CX: *C. xerosis*, LB: Luria-Bertani Broth, GAA: 30% Glacial acetic acid.

**Figure 4 antibiotics-12-00271-f004:**
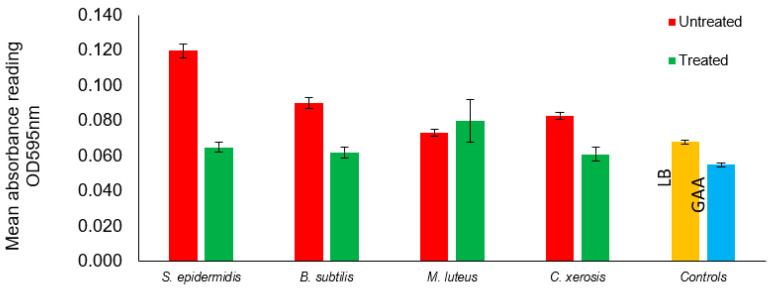
Bar graphs show inhibition of axillary microbiota biofilm by CD9 tetraspanin peptides against axillary microbiota.

**Table 1 antibiotics-12-00271-t001:** Susceptibility of CD9 tetraspanin peptide against armpit bacterial flora.

	*B. subtilis*	*S. epidermidis*	*M. luteus*	*C. xerosis*
	Inhibition Zones (mm) of Three Replicates ± SD			
CD9 peptide (1 mg/mL)	6.00 ± 0	7.67 ± 2.89	9.67 ± 4.04	7.00 ± 2.00
Vancomycin (30 μg)	19.67 ± 1.53	19.33 ± 0.58	24.33 ± 0.58	24.75 ± 20.84

**Table 2 antibiotics-12-00271-t002:** The minimum inhibitory concentration (MIC) and minimum bactericidal concentration (MBC) of CD9 tetraspanin peptides against armpit bacterial flora.

Armpit Bacterial Flora	MBC (mg/mL)	MIC (mg/mL)
*S. epidermidis*	1	0.5
*B. subtilis*	1	0.125
*M. luteus*	1	0.5
*C. xerosis*	1	0.25

**Table 3 antibiotics-12-00271-t003:** Biofilm formation of axillary microbiota with and without D9 tetraspanin peptides.

Samples	Untreated (without CD9 Peptides)	Treated (with CD9 Peptides)	% Inhibition	*p*-Value
	Mean ± SD Absorbance Reading OD 595 nm			
*S. epidermidis*	0.120 ± 0.004	0.065 ± 0.003	45.8	<0.05
*B. subtilis*	0.090 ± 0.003	0.062 ± 0.003	31.11	<0.05
*M. luteus*	0.073 ± 0.002	0.080 ± 0.012	−9.59	n.a
*C. xerosis*	0.083 ± 0.002	0.061 ± 0.004	26.51	<0.05
LB (control)	0.071 ± 0.004	0.068 ± 0.001		n.a
GA (control)	0.055 ± 0.001	0.055 ± 0.001		n.a

## Data Availability

Not applicable.

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
