# Peer review of "Antimicrobial Effects of Tetraspanin CD9 Peptide against Microbiota Causing Armpit Malodour"

_antibiotics, 2023, doi:10.3390/antibiotics12020271_

Round 1

Reviewer 1 Report

In the article, Hassanain and colleagues reported the antimicrobial effects of tetraspanins CD9 peptide against four bacterial strains causing armpit malodor. Overall, the study design is well constructed, the methods are adequately described, and the results are clearly presented.

Minor comments:
1. Line 68-69: The sentence is out of place in the context. Suggest removing it, or moving it to the discussion section.
2. Line 74-75: In the article, the author stated that the antibacterial effect of CD9 tetraspanin peptides against various bacteria is still unknown. This contradicts the discussion section where authors cited several articles which use the peptide for S. epidermis and M. Luteus.
3. Please provide references supporting the four bacterial strains as the common strains responsible for producing a foul odour.
4. There is a number of typographical, grammatical and punctuation errors which has been highlighted in the article.

Reviewer 2 Report

The aim of this work is very useful, but the experiment design and presentation are not well suited. 

1.     Authors need to include the bacterial strain number in the in the materials and methods part.

2.     In MIC determination authors should mention the range of concentrations they used. In each experiment, author should mention the concentrations of CD9 tetraspanin were used along with total volume of peptide. 

3.     The biofilm inhibition study is confusing as authors did not mention the concentrations they used. I was wondering why authors did not use the MIC doses for detecting antibiofilm activity. Also need to include the positive control (commercially available standard drug used against those bacteria). I suggest authors to show the figure with bar diagrams. For antibiofilm study author should show the percentage of inhibition. I also suggest quantifying the total protein inhibition (%) following CD9 treatment.

Reviewer 3 Report

The paper entitled "Antimicrobial Effects of Tetraspanins CD9 Peptide Against Microbiota Causing Armpit Malodor" is a short but well prepared study showing the applicability of a synthetic peptide CD9 against skin microbiota. I have only few minor issues for the Authors to consider:

1. MBC and MIC values are quite high. How does it affect the potential applicability of peptides as additives in hygienic products? What is expected cost-to-benefit ratio? How expensive would be the production of peptides?

2. Discussion appears to be a bit messy and should be reorganised. What stands out is the fact that the general description of peptides is provided twice, in lines 172-178 and 189-191. I would like to ask the Authors to read the Discussion section once again and try to rewrite it in a more organized way.

3. Sometimes the Authors miss the units (lines 86-88)

4. In line 234 the Authors wrote about previous studies without mentioning the exact reference.

Round 2

Reviewer 2 Report

Thank you for addressing my questions. I suggest the work for publication.